# Optimization of Nutrient-Rich Ice Plant (*Mesembryanthemum crystallinum* L.) Paste Fresh Noodle Pasta Using Response Surface Methodology

**DOI:** 10.3390/foods12132482

**Published:** 2023-06-25

**Authors:** Yeo-Wool Kang, Na-Mi Joo

**Affiliations:** Department of Food and Nutrition, Sookmyung Women’s University, Cheongpa-ro 47gil 100, Yongsan-gu, Seoul 04310, Republic of Korea; fooddock@sookmyung.ac.kr

**Keywords:** ice plant, paste, raw pasta, RSM

## Abstract

The ice plant is a species that is grown mainly in the dry regions of the American West and contains various minerals and ingredients beneficial for human health, such as inositol and beta-carotene. With the growing trend towards healthy foods, pasta consumption has also increased. Pasta is a convenient and low-glycemic-index food that is composed mainly of carbohydrates, proteins, lipids, dietary fiber, and trace amounts of minerals. The optimal mixing ratio was evaluated to produce pasta of the highest quality in terms of blood sugar elevation and antioxidant efficacy. The components and minerals of the ice plant, including D-pinitol and inositol, were analyzed, and 20 essential amino acids were identified. In this study, we also investigated the quality and characteristics of ice plant paste and eggs, as well as the quality, antioxidant activity, and formulation of raw materials mixed with ice plant at different ratios. Optimal conditions were found to be 46.73 g of ice plant paste in 100 g of durum wheat flour, 20.23 g of egg, and 2 g of salt, providing a way to develop fresh pasta that enhances the health benefits of ice plant paste without excessive moisture and other ingredients.

## 1. Introduction

The ice plant is an annual plant belonging to the genus Aizoaceae. The bladder cells, which look like ice crystals on the surface of stems and leaves, contain ingredients useful for the human body, such as inositol, beta-carotene, as well as various minerals. The ice plant is reported to exhibit various physiological activities, such as antioxidant properties, blood-sugar-lowering activities, and antibacterial effects [1]. It is mainly cultivated in the dry regions of the west of America and is called ice plant because the shape of the salt released from the bladder cells, which resembles ice crystals [2]. In Africa, it has traditionally been consumed to prevent dehydration by replenishing water and salt, and in Europe and Japan, it has been spotlighted as a highly functional vegetable that has both salty and savory tastes and has been applied to various foods for decades [2]. As the various effects of ice plants have become known, the number of farms cultivating them in large quantities, in addition to consumption, has increased in Korea. Despite its efficacy in the treatment of diabetes, hypertension, and obesity, since ice plant extract prevents lipogenesis and increases lipolysis, ice plant research in Korea is still insufficient [3]. The ice plant exhibits a wide range of activities, including antioxidant properties; radical scavenging activity [4,5]; and antihypertensive, hypoglycemic, hyperglycemic, nootropic [6], and memory-improving activity [7]. It has been reported that ice plant (*Mesembryanthemum crystallinum*) extract ameliorates hyperglycemia [8]. Other studies have shown that the ice plant bladder cells contain polyols, such as D-pinitol, inositol, and myo-inositol, that show antioxidant activity and, in the case of D-pinitol, blood sugar reduction [9,10,11]. In recent years, an eating-out culture centered on Western cuisine has developed rapidly worldwide. Therefore, pasta consumption have also increased rapidly. Increasing demand by a growing number of health-conscious consumers for healthy foods has garnered interest from food manufacturers, and a plethora of studies have been published in the literature to date [12,13]. As the primary source of energy for humans, carbohydrates are found in food and play a crucial role in the human diet. Pasta, a food that is well-known globally for its convenience, affordability, and low glycemic index (GI), is composed mainly of carbohydrates (70–76%), proteins (10–14%), lipids (1.8%), dietary fiber (2.9%), and trace amounts of minerals and vitamins [14]. Made from semolina flour from either durum wheat or common wheat, pasta is created by kneading flour with water (28–32% *w*/*w*) and utilizing mechanical force to produce fresh pasta, which is then dried. Despite its high carbohydrate content, pasta contains low levels of dietary fiber, vitamins, essential amino acids, and minerals, some of which are lost during the milling process. However, the FDA recognizes the presence of bioactive ingredients in pasta, such as proteins, minerals, phytochemicals, vitamins, etc., with up to 10–15% of these functional ingredients retained without a significant decrease in pasta quality.

Pasta is a typical Italian noodle dish and is divided into raw and dried pasta. In Korea, dried pasta is the most commonly used form, while in the western and northern regions of Italy, fresh raw pasta is preferred due to its mild flavor and the ability to enhance its color, shape, and nutritional value with added ingredients. The trend towards raw noodles is growing, with changes in consumer preferences leading to alterations in dietary habits. Different types of pasta, including yellow lentil pasta [15]; pasta fortified with oat bran and apple flour [16]; onion skin powder wheat pasta [17]; olive pomace pasta [18]; chickpea flour and protein isolate pasta [19]; *Cistus incanus* L. leaves on wheat pasta [20]; add cricket powder pasta [21]; and pasta enriched with ingredients such hemp seeds [22], spirulina [23], and Moringa oleifera leaf powder [24], have been reported, but there is still room for improvement in nutritional content. In this study, the amount of ice plant paste added to pasta better exploits its positive physiological functions. After manufacturing pasta and developing raw pasta with improved health benefits and consumer appeal, the optimal mixing ratio was evaluated to produce pasta of the highest quality in terms of blood sugar elevation, antioxidant efficacy, and raw pasta quality.

## 2. Materials and Methods

### 2.1. Raw Materials

The ice plant used in this study was purchased in October 2022 from Yesan ice plant Signal Agricultural Corporation (Yesan, Republic of Korea) in Chungcheongnam-do. The general components, inorganic components, inositol, and amine were analyzed in each part of the ice plant. After analysis, durum wheat (Divella Co., Rutigliano, Italy), salt (CJ Che-iljedang Co., Seoul, Republic of Korea), and eggs (Pulmuone Co., Eumseong, Republic of Korea) were purchased and used as ingredients for pasta manufacturing. DW (distilled water) and 70% ethanol were used as the extraction solvents in the antidiabetic and antioxidant experiments. The reagents used in this experiment were products of Sigma-Aldrich Co. (St. Louis, MO, USA).

### 2.2. Proximate

The quantification of raw fat, crude protein ash, and carbohydrates in each part of the ice plant was performed by general component analysis using the AOAC [25] method. Crude fat was measured according to the Soxhlet extraction method using an automatic crude fat extractor (Soxtec Avanti 2050, FOSS Co., Hillerod, Denmark). Crude protein was measured by the micro-Kjeldahl method using an automatic nitrogen distillation apparatus (Kjeltec Analyzer 2300, FOSS Co., Hillerod, Denmark), and crude fiber was analyzed by the Henneberg--Stohman method. Ash content was analyzed by direct ash analysis at 550 °C (JSMF-140T, JSR Co., Seoul, Republic of Korea). Each experiment was repeated three times, and the results are expressed as average values. The carbohydrate content is the value excluding moisture, ash, crude protein, and crude fat.

### 2.3. Mineral contents

The raw materials contained in each part of ice plant (*M.crystallinum* L.) are calcium, iron, potassium, sodium, magnesium, manganese, zinc, copper, cadmium, and lead, which are approved by the Ministry of Food and Drug Safety (2016) [26]. A volume of 8 mL of 70% nitric acid (Dongwoo Co., Seoul, Republic of Korea) and 2 mL of distilled water were added to about 1.0 g of the sample, and the nitric acid was decomposed in a microwave digester (Multiwave ECO, Anton-paar, Graz, Austria) with 50 mL distilled water. Inorganic substances such as calcium, iron, potassium, sodium, magnesium, manganese, and zinc were used, and an inductively coupled plasma optical emission spectrometer (ICP-OES, ACTIVA, Horiba Jobin Yvon, Villeneuve-d’Ascq, France) was used. Trace minerals such as copper, cadmium, and lead were analyzed with an inductively coupled plasma mass spectrometer (ICP-MS, Agilent 7700 ICP/MS.Agilent).

### 2.4. D-Pinitol and Inositol Content 

After homogenizing the raw sample for each part of the ice plant (*M. crystallinum* L.), about 1 g was taken, and 50 mL of a 5 mM sodium 1-hexanesulfonate solution was added. Thereafter, the extract was extracted for 30 min with an ultrasonic extractor (JAC 4020, Kodogiyeon, Hwaseong, Republic of Korea) at 40 °C, mixed well with the sample, and centrifuged at 15,000 rpm for 15 min. Next, only the supernatant was taken and filtered with a 0.45 μm syringe filter (Whatman Inc., Maidstone, UK), and HPLC (Hitachi 5000 chromaster, Hitachi Ltd., Tokyo, Japan) analysis was performed.

### 2.5. Amino Acid Content Analysis

A Dionex Ultimate 3000 HPLC instrument (Thermo Dionex, Waltham, MA, USA) and an Agilent 1260 infinity FL detector (Agilent, Santa Clara, CA, USA) were used for analysis. The conditions for use were as follows. The FL detector settings consisted of an emission wavelength of 450 nm and an excitation wavelength of 340 nm (OPA). Additionally, the FL detector had an emission wavelength of 305 nm and an excitation wavelength of 266 nm (FMOC). A UV detector was used at 338 nm, and as a standard, 17 types of 1 nmol/uL amino acids were dissolved in 0.1 N-HC and diluted with tertiary distilled water to test at concentrations of 1000, 500, 100, and 10 pmol/uL. For the analysis conditions, an Inno C18 column (4.6 mm × 150 mm, 5 um/Youngjin Biochrom, Republic of Korea) was used, and mobile phase A was 40 mM Sodium phosphate, with a pH value of 7. Mobile phase B was 3DW/acetonitrile/methanol (10: 45: 45 *v*/*v*%). The column temperature was 40 ℃and the sample temperature was 20 °C. A sample volume of 0.5 uL was injected for sufficient ionization of the material to be analyzed.

### 2.6. Total Dietary Fiber Content

The raw samples of ice plant (*M.crystallinum* L.) parts were analyzed using the a Fibertec 1023 System E dietary fiber analyzer (FOSS, Denmark). The process involved adding 50 mL of 0.08 M phosphate buffer (pH 6.0) to 1 g of the sample, then adding 0.1 mL of α-amylase solution and performing hydrolysis at 95–100 °C for 15 min. After cooling to room temperature, 10 mL of 0.275 N sodium hydroxide solution was added to adjust the pH to 7.5, followed by the addition of 0.1 mL of protease and hydrolysis in a shaking water bath at 60 °C for 30 min. The solution was cooled again, and 10 mL of 0.325 M HCl solution was added to adjust the pH to 4.0–4.6. Then, 0.1 mL of amyloglucosidase was added, hydrolyzed in a 60 °C shaking water bath for 30 min, and cooled to room temperature. Subsequently, 95% ethanol (equivalent to four times the amount of the hydrolyzate) was added, and the mixture was stirred and left to stand at room temperature for a day. Finally, celite was added, the mixture was filtered through a crucible to obtain a constant weight; washed with 78% ethanol, 95% ethanol, and acetone in sequence; dried in a hot air dryer at 60 °C for 24 h; and weighed to calculate the results using the following formula:Total dietary fiber (%) = {(residue weight after drying − ash amount− protein amount − blank)/(sample weight)} × 100(1)

### 2.7. α-Glucosidase Inhibitory Activity

To measure the α-glucosidase inhibitory activity of ethanol for each part of the ice plant, the procedure described in [27] was applied. α-Glucosidase obtained from yeast (Sigma-Aldrich Co., St, Louis, MO, USA) was used as the enzyme, and the substrate measured the production of ρ-nitrophenol using ρ-nitrophenol-α-glucopyranoside. A volume of 2 mL of the sample was added to 100 μL of α-glucosidase and incubated at 37 °C for 10 min. Next, 200 μL of 10 mM ρ-nitrophenol-α-glucopyranoside was added, stirred, and reacted at room temperature for 20 min. After stopping the reaction by adding 5 mL of 1 N sodium hydroxide (Ducksan Pure Chemicals, Seoul, Republic of Korea), 5.6 mL of 50 mM phosphate buffer was added (pH 6.8), absorbance was measured at 405 nm, and the value was calculated using the following formula:α-glucosidase inhibition activity (%) = (1 − [A/B]) × 100(A: Experiment, B: Control)(2)

The α-glucosidase inhibitory activity of fresh raw pasta with ice plant paste was measured in the same manner as the measurement of α-glucosidase inhibitory activity of ice plant.

### 2.8. α-Amylase Inhibitory Activity

The α-amylase inhibitory activity was evaluated by applying the procedure described in [28]. The inhibitory activity of pancreatin-derived α-amylase was measured using starch as a substrate. After stirring α-amylase (Sigma-Aldrich Co., St, Louis, MO, USA) in 0.02 M phosphate buffer (Ducksan Pure Chemicals, Seoul, Republic of Korea) with a pH of 6.9 and leaving it at 37 °C for 10 min, 1% starch was added. The sample was then incubated for 10 min at 37 °C as a substrate. Dinitrosalicylic acid (Sigma-Aldrich Co., St, Louis, MO, USA) was added, and the reaction was stopped by heating at 95° C for 5 min; then, distilled water was added, and absorbance was measured at 540 nm.
α-amylase inhibition activity (%) = (1 − [A/B]) × 100(A: Experiment, B: Control)(3)

The α-amylase inhibitory activity of fresh noodle pasta with ice plant paste was measured in the same manner as the measurement of α-amylase inhibitory activity of ice plant.

### 2.9. Fresh Noodle Pasta Preparation and Optimization

Materials including an ice plant (purchased in October 2022 from Shinho Agricultural Co. in Yesan, Republic of Korea), durum wheat (from Divella Co. in Rutiliano, Italy), salt from CJ (CheilJedang Co. in Seoul, Republic of Korea), and eggs (from Pulmuone Co. in Seoul, Republic of Korea) were used for experimentation. Fresh ice plant paste was used for raw pasta production. When making pasta, the ice plant was ground using its natural water, then pacotized and freeze-dried to a moisture content of 30% to make a paste. The optimal mixing ratio of ice plant paste and eggs was determined using the Design-Expert program. The experiments were designed using central composite design (CCD) and response surface design (RSM) to evaluate various sensory characteristics, such as chromaticity (L = lightness, a = redness, and b = yellowness), texture, and overall quality. The ice plant paste was set at 30–60 g, and the egg was set at 10–30 g, with the mixing ratio adjusted through a preliminary test (Table 1). All ingredients were mixed in a food mixer, hand-kneaded, and stored in a refrigerator. The dough was then rolled out and cut into raw pasta using a manual pasta machine. The resulting fresh pasta was 4 mm wide, 1 mm thick, and 300 mm long.

#### 2.9.1. Color Measurement

To determine the chromaticity of each part of the ice plant, the values of L (lightness), a (redness), and b (yellowness) were measured using a colorimeter (Chroma Meter CR-300, Minolta Co. Osaka, Japan). Each of the three samples for each part was measured three times. The L value of the standard white plate used at this time was 105.61, the a value was −0.18, and the b value was +4.82.

#### 2.9.2. PH Measurement

The pH measurement for each part of the ice plant was homogenized by adding 50 mL of distilled water to 5 g of the sample. A glass electrode pH meter (F-51, HORIBA, Tokyo, Japan) was used to measure three samples for each part, with each sample measured three times. The average value was obtained.

#### 2.9.3. Salt Meter Measurement

To measure the salinity of each part of the ice plant, samples were homogenized by adding 50 mL of distilled water to 5 g of the sample. Then, 100 mL of distilled water was used for salinity analysis, and 10 mL of the homogenized mixture was transferred to a test container. The sample was zeroed using a 0.3% NaCl solution before analysis. The titration reagent was titrated with 0.1 N AgNO_3_ (Daejung, Chemical & Metals Co., Ltd., Siheung-si, Republic of Korea). The analysis was performed using a salinity meter (Metrohm USB, sample processor, 785 DMP Titrino, Metrohm AG, Herisau, Switzerland).

#### 2.9.4. Moisture Content Measurement

The moisture content of fresh noodle pasta with ice plant paste was evaluated using the atmospheric pressure drying method at 105 °C. The evaluation was conducted using a dry oven (SW-90D, Sanwoo, Seoul, Republic of Korea). Prior to cooking, 0.5 g of raw pasta was weighed and placed in an aluminum dish. The dish was then heated to 105 °C, and the moisture content was measured. After multiple measurements were taken, the average value was calculated and presented.

#### 2.9.5. DPPH, Total Polyphenol, and Total Flavonoid Antioxidant Contents

The sample solution was prepared by mixing 70% ethyl alcohol with 5 g of raw ice plant paste noodle pasta before cooking. Then, 50 mL of the solution was added to a stirring device (Bagmixer 400w, Interscience, St. Nom, France) and homogenized for 90 s. The mixture was then placed in a shaking incubator (24 ℃, 100 rpm) for 24 h. The sample was obtained by filtering the mixture twice using Whatman No. 4 filter paper.

(A)Total Phenol Content

The total phenolic content was determined by modifying the Folin–Denis method (1912) and measured using the Singleton–Rossi method (1965). Volumes of 2400 μL of distilled water and 150 μL of the sample were mixed in a test tube. Then, 150 μL of 2 N Folin–Ciocalteu phenol reagent (Sigma-Aldrich Co., St. Louis, MO, USA) was added and stirred. Then, 1 N Na_2_CO_3_ was added, and the mixture was reacted for 2 h in the dark. The absorbance at 725 nm was measured using a UV-visible Spectrophotometer (V-530, Jasco Co., Tokyo, Japan). Gallic acid was used as the standard and analyzed in the same manner as the sample. A calibration curve was established, and the total phenol content of the sample was obtained as milligrams of gallic acid equivalent (GAE) per dry basis.

(B)Total Flavonoid Content

The total flavonoid content was determined by modifying the Davis method [29]. First, 1 mL of the sample was mixed with 10 mL of diethylene glycol (Junsei Chemical, Tokyo, Japan) and 1 mL of 1 N sodium hydroxide (Ducksan Pure Chemicals, Seoul, Republic of Korea). The mixture was strongly stirred and reacted in a 37 °C water bath for 60 min. The absorbance was measured at 420 nm using a spectrophotometer. Catechin was used as the standard and analyzed in the same manner as the sample. A calibration curve was established, and the total flavonoid content of the sample was obtained as milligrams of catechin equivalent (CE) per dry basis.

(C)DPPH Free Radical Scavenging Activity

The free radical scavenging activity was determined using the method described by [30]. A DPPH solution (1.5 × 10^−4^ M, final concentration) was mixed with 0.5 mL of the sample, stirred, and brought to room temperature. After 30 min, the absorbance was measured at 517 nm using a UV-visible spectrophotometer (V-530, Jasco Co., Tokyo, Japan). The DPPH free radical scavenging activity value was obtained using the following formula:DPPH radical scavenging activity (%) = (1 − A/B) × 100(A: Experiment, B: Control)(4)

#### 2.9.6. Sensory Analysis

For a sensory evaluation study of raw pasta using ice plant paste, we used a 7-point hedonic scale to assess the participants’ preferences, with 1 indicating a strong dislike and 7 indicating a favorable opinion. The panel of 20 participants was composed of individuals with experience in food tasting capable of differentiating various sensory attributes, such as taste, flavor, and texture. The panelists were between the ages of 20 to 50 years old, and the samples were presented in a random order and coded to ensure anonymity. The panelists were each given the opportunity to evaluate the samples, with taste-neutral water provided for rinsing between samples, and all samples were presented at room temperature.

#### 2.9.7. Statistical Analysis

The results of the sensory evaluation study were analyzed statistically using the central composite design (CCD) and the response surface methodology (RSM). The data were analyzed using quadratic analysis of variance (ANOVA) to evaluate each response variable and determine the influence of various factors on the outcome. To compare the means between the control and optimized raw pasta of ice plant paste, ANOVA was performed using Microsoft Excel at a significance level of f-value 0.05.

## 3. Results

### 3.1. General Component Analysis of Ice Plant

In the context of a food products, a general component could refer to the basic ingredients or nutrients that make up the product, such as protein, carbohydrates, fats, vitamins, and minerals. Table 2 shows the analysis results for moisture, crude protein, crude fat, crude ash, and carbohydrate contents in ice plants. These results indicate that there were significant differences in the composition of the ice plant depending on the part of the plant. When compared to other halophytes, the ice plant has a lower ash (0.87–1.99%) and crude fat (0.03–0.13%) contents than those of glasswort and ash plants. For example, saltwort plants were found to contain 17.76% crude protein, 10.11% moisture, 20.38% ash, and 2.05% crude fat, while general ingredients of seaweed were identified as 12.30% crude protein, 5.10% moisture, 46.80% ash, and 1.30% crude fat.

### 3.2. Mineral Content

Macro minerals are minerals that are required in relatively large amounts by the human body for normal physiological function. They include calcium, phosphorus, magnesium, sodium, potassium, chloride, and sulfur. These minerals are essential for many bodily processes, such as bone health, muscle and nerve function, fluid balance, and acid–base balance. It is important to maintain a balanced intake of these macro minerals through a healthy diet to support optimal health and prevent deficiencies or imbalances (Table 3). Microminerals, also known as trace minerals, are essential minerals that the body requires in very small amounts—generally less than 100 mg per day. They include iron, zinc, copper, manganese, iodine, fluoride, chromium, molybdenum, selenium, and cobalt. Despite their small amounts, they play critical roles in various physiological processes, such as enzyme function, immune system function, growth and development, and energy metabolism. A balanced intake of microminerals is necessary for optimal health, and deficiencies or imbalances in these minerals can lead to a variety of health problems [31]. Table 3 presents the results of the analysis of calcium, iron, potassium, sodium, magnesium, manganese, zinc, and copper contents in each part of the ice plant. In addition to sodium, potassium, magnesium, and calcium, which are normally abundant in halophytes, iron, manganese, copper, and zinc were also found to be abundant in the ice plant. As expected from the salty taste of the ice plant, the highest content was sodium, followed by potassium and magnesium. Among the trace minerals, manganese content was the highest, and the contents of copper, manganese, and zinc were also higher than those of other terrestrial plants. In a study of halophytes [32], namunjae was reported to have higher sodium, potassium, and calcium contents than glasswort but a slightly lower iron content, while manganese, copper, and zinc contents were higher than those of glasswort. Compared to saltwort and seaweed, the ice plant was found to have higher contents of mineral components, including sodium, potassium, calcium, and magnesium.

### 3.3. Cytosol Content Analysis

#### 3.3.1. Pinitol Content Analysis in Different Parts

Pinitol is a cyclic polyol (also known as 3-O-methyl-chiro-inositol) that occurs naturally in plants and microorganisms. Its chemical structure is similar to that of inositol: a six-carbon cyclic polyol that plays a vital role in several physiological processes. Derived from various plants and microorganisms, pinitol is a cyclitol that is sometimes referred to as 3-O-methyl-chiro-inositol. It shares a structural resemblance to inositol, a cyclic polyol with six carbons that is present in many living organisms and is essential for multiple biological functions [33]. Pinitol is one of the many types of cyclic polyols that occur naturally in plants and microorganisms and is also known as 3-O-methyl-chiro-inositol. Similar in structure to inositol, pinitol has potential health benefits, such as anti-inflammatory, antioxidant, and antidiabetic properties. Additionally, its sweet taste makes it a promising non-caloric sweetener that could find use in the food and pharmaceutical industries. Pinitol can be found in a variety of plant sources, including pine trees, carob trees, soybeans, and certain fruits and vegetables.

#### 3.3.2. Inositol Content Analysis in Different Parts

Inositol, a type of cyclic polyol or sugar alcohol, is crucial for numerous physiological processes in living organisms due to its structural similarity to glucose. Inositol exists in many different forms, with myo-inositol being the most common and most widely studied in humans. Inositols occur naturally in many foods, such as fruits, beans, and grains, and are also produced by the body. They serve vital functions in various cellular signaling pathways, including those that regulate cell growth, differentiation, and death, as well as those involved in insulin signaling, lipid metabolism, and neurotransmission [34]. Furthermore, inositols have been the subject of research, owing to their possible therapeutic effects, particularly in the treatment of conditions such as polycystic ovary syndrome, metabolic syndrome, and anxiety disorders.

In this study, we present the analysis results of inositol components in the ice plant (Table 4). Inositol composition was found to be highest in the stem (44.45 ± 1.44 mg/100 g), followed by the cotyledon (10.53 ± 0.47 mg/100 g) and net (4.36 ± 0.25 mg/100 g). HPLC analysis of the ice plant also revealed the presence of pinitol and inositol in each part of *M. crystallinum* L. (as shown in the chromatogram), the respective contents of which are presented in Table 3. Pinitol is a unique inositol component primarily found in pine needles, soybeans, and carob [35]. In the 1990s, researchers at the University of Virginia discovered that chiro-inositol, a form primarily present in plants, had a hypoglycemic effect similar to that of pinitol [36,37]. Since then, the blood-sugar-enhancing effect of pinitol has been confirmed through animal and clinical trials in several studies, leading to further interest in blood-sugar-enhancing materials [38].

### 3.4. Amino Acid Content in Different Parts

Our analysis revealed the presence of a total of 20 amino acids (Table 5). Aspartic acid is a non-essential amino acid used in protein biosynthesis and the citric acid cycle. It can be synthesized in the body or obtained from dietary sources. Glutamic acid is a non-essential amino acid used in protein synthesis, energy production, acid–base balance, transamination, and glutamate metabolism. It can be produced by the body or obtained from dietary sources. Both aspartic acid and glutamic acid are important for overall health and well-being and are involved in numerous metabolic processes in the body. Asparagine is a non-essential amino acid that is used by the body to build proteins. It is also involved in the regulation of the acid–base balance and the metabolism of nitrogen, among other functions. Serine is a non-essential amino acid that is used by the body to build proteins. It is also involved in the metabolism of fatty acids and is a precursor for several important biomolecules, including glycine and glutathione. Glutamine is a non-essential amino acid that plays a crucial role in protein synthesis, as well as energy production, nitrogen metabolism, acid–base balance, and neurotransmitter function. Histidine is an essential amino acid that is important for protein synthesis, acid–base balance regulation, neurotransmitter synthesis, and detoxification. Glycine is a non-essential amino acid that plays a role in protein synthesis, neurotransmitter function, acid–base balance regulation, hemoglobin synthesis, and biomolecular synthesis. Threonine is an essential amino acid that is involved in protein synthesis and the formation of collagen and elastin. It is also involved in the metabolism of lipids and is a precursor for the synthesis of other important biomolecules. Arginine is a semi-essential amino acid that is involved in protein synthesis, the regulation of blood flow, and the immune response. It is a precursor for the production of nitric oxide, is involved in the metabolism of urea, and is a source of energy for cells.

Alanine is a non-essential amino acid that is involved in protein synthesis and the regulation of blood sugar levels and is a source of energy for cells, particularly muscle cells. It is also involved in the metabolism of nitrogen [39].

GABA is a neurotransmitter that is involved in the regulation of nerve impulses in the brain. It is synthesized from the amino acid glutamate and has been linked to the regulation of mood and anxiety [40].

Tyrosine is a non-essential amino acid that is involved in protein synthesis and the synthesis of several important neurotransmitters. It is also involved in the regulation of mood and is believed to play a role in the regulation of stress and anxiety [41].

Valine, methionine, tryptophan, phenylalanine, isoleucine, and leucine are amino acids that are considered essential because the body cannot produce them on its own and must obtain them from dietary sources [42]. They are used by the body to build proteins involved in many important functions in the body, such as enzymatic activity, hormone regulation, and structural support. In addition, each amino acid has its own specific functions and metabolic pathways, such as blood sugar regulation; energy production; tissue maintenance and repair; neurotransmitter and hormone synthesis; and regulation of mood, sleep, and appetite, as well as skin, hair, and nail health. Together, these essential amino acids help to maintain overall health and well-being. Lysine and proline are amino acids that serve vital functions in the body. Lysine, an essential amino acid, is involved in numerous processes, such as protein synthesis, calcium absorption, collagen formation, and carnitine production [43]. Proline, a non-essential amino acid, also plays a significant role in protein synthesis and collagen formation, in addition to regulating blood pressure and maintaining healthy skin, hair, and nails [44]. The combination of these two amino acids is essential to maintaining optimal health and well-being.

### 3.5. Total Dietary Fiber Content

Total dietary fiber refers to the indigestible components of plant-based foods that resist digestion and absorption in the small intestine. Table 6 shows the total dietary fiber for each part of the ice plant. These include non-starch polysaccharides such as cellulose, hemicellulose, and pectin, as well as other substances such as resistant starch and lignin. Adequate intake of total dietary fiber is essential for maintaining a healthy digestive system; promoting feelings of fullness; and reducing the risk of chronic diseases, such as heart disease, diabetes, and certain cancers. The recommended daily intake of total dietary fiber varies based on factors such as age and gender but generally falls between 25 and 38 g per day for adults [45].

### 3.6. α-Glucosidase Inhibitory Activity and α-Amylase Inhibitory Activity

α-Glucosidase inhibitory activity and α-amylase inhibitory activity refer to two methods of measuring how well a substance can hinder the activity of enzymes responsible for carbohydrate digestion. α-Glucosidase is an enzyme that breaks down complex carbohydrates into simple sugars, while α-amylase is an enzyme that breaks down starches into smaller carbohydrate molecules. Limiting the activity of these enzymes can result in slower carbohydrate absorption, potentially aiding in the management of conditions such as diabetes and obesity. Researchers often investigate substances that show α-glucosidase inhibitory activity or α-amylase inhibitory activity to explore their potential as natural remedies or dietary supplements.

#### 3.6.1. α-Glucosidase Inhibitory Activity in Different Parts

α-Glucosidase is an enzyme that converts oligosaccharides or disaccharides into glucose. α-Glucosidase inhibitors delay the digestion and absorption of carbohydrates to control postprandial blood sugar without causing hyperinsulinemia or hypoglycemia, which can lead to obesity, and promote insulin secretion. They also promote the secretion of glucagon-like peptide-1, which inhibits glucagon secretion in the small intestine [46]. The α-glucosidase and α-amylase inhibitory activities of the ethanol extract from different parts (stem, cotyledon, and shoot) of *M. crystallinum* L. were investigated at a concentration of 1 mg/mL, as shown in Table 7. Carbohydrates are broken down into disaccharides by α-amylase, which are then broken down into monosaccharides by α-glucosidase, a disaccharide-digesting enzyme found in the brush border membrane of small intestine epithelial cells. When α-glucosidase is inhibited, glucose absorption in the small intestine is slowed down, which helps to control blood glucose levels by preventing a rapid increase in blood sugar after meals. Stem extract at 62.5 ppm exhibited 3.81 ± 0.34% inhibition, while 125 ppm, 250 ppm, 500 ppm, and 1000 ppm resulted in 5.33 ± 0.67%, 8.84 ± 0.35%, 15.55 ± 0.24%, and 25.45 ± 0.55% inhibition, respectively, with an IC_50_ value of 7995 ppm. The cotyledon extract exhibited 5.99 ± 0.53% inhibition, with an IC_50_ value of 5117 ppm. In the case of shoot extract, inhibition increased significantly from 125 ppm to 5.52 ± 0.59% at 1000 ppm, with an IC_50_ value of 4592 ppm. At the highest concentration of 1000 ppm, α-glucosidase activity was inhibited in the order of stem > cotyledon > shoot.

#### 3.6.2. α-Amylase Inhibitory Activity in Different Parts

α-Amylase is an enzyme that breaks down the α-D-(1,4)-glucan bond in carbohydrates. When α-amylase inhibitors are present, the digestion of starch in the small intestine is slowed down, which delays the absorption of glucose and prevents a rapid increase in blood sugar after meals. Both salivary and pancreatic α-amylase are essential enzymes for carbohydrate metabolism in various living organisms. The α-amylase activity inhibition ability of ethanol extracts from different parts of *M. crystallinum* L. was measured in this study. The stem extract showed concentration-dependent inhibition, with 7.44 ± 1.16% inhibition at 62.5 ppm and 31.91 ± 3.90% inhibition at 1000 ppm, with an IC_50_ of 2451 ppm. The cotyledon extract also showed concentration-dependent inhibition, with 1.58 ± 0.82% inhibition at 62.5 ppm and 17.55 ± 1.69% inhibition at 1000 ppm, with an IC_50_ of 4748 ppm. The shoot extract showed 2.06 ± 1.10% inhibition at 62.5 ppm and 23.04 ± 1.55% inhibition at 1000 ppm, with an IC_50_ of 7115 ppm. At the highest concentration of 1000 ppm, the order of inhibition was stem > shoot > cotyledon for all three ethanol extracts (Table 7).

### 3.7. Quality Characteristics of Fresh Noodle Pasta Using Response Surface Analysis

Table 8 shows the physicochemical properties of fresh noodle pasta with ice plant paste added. With respect to the chromaticity of pasta, the L, a, and b values ranged from 30.59 to 36.19, −14.08 to −11.61, and 16.38 to 19.99, respectively, and a linear model was selected for each color component. The pH of the pasta ranged from 6.16 to 6.47, and a linear model was selected in which the amount of eggs had a greater effect than the amount of ice plant (Figure 1). Salinity ranged from 0.26 to 0.34%, and a linear model with a significant *p*-value of 0.004 and an R^2^ value of 0.96 was selected (Table 9). Moisture content ranged from 10.91 to 14.41%, and a two-order interaction model between ice plant paste and egg yolk was selected, with a significant *p*-value and an R^2^ value of 0.84 (Table 9). The amount of both ice plant paste and egg yolk affected the moisture content of the pasta.

### 3.8. Antioxidation

The DPPH radical scavenging ability of raw pasta increased as the amount of added ice plant paste increased, with a range of 75.90–89.80% (Table 10). A linear model was selected, and the R^2^ value was 0.903, indicating high reliability. The perturbation plot and response surface plot presented in Figure 2 show that the addition of ice plant paste increased the DPPH radical scavenging ability, while the amount of eggs had no effect. Similar results were reported in a study by Hwang et al. (2011) [47], in which the DPPH radical scavenging activity increased with the concentration of Cheongyang red pepper juice.

The total phenol content of fresh pasta with ice plant paste ranged from 351.88 to 439.22 mg GAE/g, and a linear model was selected for this measurement. The *p*-value was significant at 0.0001, and the R2 value was 0.912, indicating the suitability of the model (Table 11). The addition of ice plant paste tended to increase the total phenol content (Figure 2), compensating for the reduction in antioxidant properties that occurs during the milling process of wheat pasta, at which point the phenolic components are removed [48].

The total flavonoid content of raw pasta with ice plant paste ranged from 231.08 to 470.25 mg QE/g, and a linear model was selected for this measurement (Table 10). The *p*-value was significant at 0.0022 (*p* < 0.05), and the R^2^ value was 0.921, indicating the suitability of the model (Table 11). As the amount of added ice plant paste increased, the total flavonoid content tended to increase, similar to the results reported in a study by Kim et al. (2013) [49], in which the total flavonoid content increased with the addition of buckwheat flour.

### 3.9. Starch-Degrading Enzyme Activity Inhibition

The α-glucosidase inhibitory activity of fresh noodle pasta with ice plant paste ranged from 7.25% to 14.24%, as shown in Table 12. The relationship between ice plant paste and egg factors in the context of an interacting quadratic model was selected for the α-glucosidase inhibitory activity, with a significant *p*-value of <0.05 and an R^2^ value of 0.891, indicating high reliability (Table 13). The results suggest that the addition of ice plant paste has a positive effect on α-glucosidase inhibitory activity. Ice plant paste was found to have a greater effect on the increase in α-glucosidase inhibitory activity. The perturbation plot and response surface plot presented in Figure 3 confirm the effect of the addition of ice plant paste on the α-glucosidase inhibitory activity.

The α-amylase inhibitory activity of the raw ice plant paste pasta ranged from 99.03% to 100%, as shown in Table 12. A relationship between ice plant paste and egg factors in the context of an interacting quadratic model was selected for the α-amylase inhibitory activity, with a non-significant *p*-value of 0.123 and an R^2^ value of 0.761. The results suggest that ice plant paste has a greater effect on α-amylase inhibitory activity than eggs (Table 13, Figure 3). The perturbation plot and response surface plot presented in Figure 3 confirm the effect of the addition of ice plant paste on the α-amylase inhibitory activity. We believe that ice plant paste directly affects the α-amylase inhibitory activity.

### 3.10. Sensory Characteristics

The sensory characteristics of fresh ice plant paste pastas were evaluated for color, flavor, taste, texture, appearance, and overall acceptability using a seven-point preference scale (Table 14). The acceptability of color ranged from 3.63 to 6.50, and a quadratic model in which ice plant paste and eggs interacted was selected, showing a significant result with a *p*-value of 0.0204 (*p* < 0.05) and an R^2^ value of 0.862 (Table 15). The preference for flavor ranged from 3.31 to 5.88, and a quadratic model in which the two factors interacted was selected, with a *p*-value of 0.1131 and an R^2^ value of 0.701 (Table 15). The taste preference ranged from 3.31 to 5.63, and a quadratic model in which the two factors interacted was selected, with a *p*-value of 0.0278 (*p* < 0.05) and an R^2^ value of 0.840. The preference for texture ranged from 3.63 to 6.31, and a quadratic model in which each factor interacted was selected, with a *p*-value of 0.0234 (*p* < 0.05) and an R^2^ value of 0.790 (Table 15). The acceptability of appearance ranged from 3.63 to 6.19, and a quadratic model in which the two factors interacted was selected, with a *p*-value of 0.0500 (*p* < 0.05) and an R^2^ value of 0.801 (Table 15). The results suggest that ice plant paste has a greater effect on the sensory characteristics of the pasta than eggs. As the amount of ice plant paste and eggs increased, the degree of preference for each sensory characteristic increased, then decreased after the center point, suggesting that the blending ratio of the ice plant paste and eggs may have affected the sensory characteristics of the pasta (Figure 4).

The optimization of fresh ice plant paste pasta production yielded significant results as sensory test items including color, taste, flavor, texture, appearance, and overall quality were maximized within the range of independent variables of ice plant paste and eggs. A canonical model was used to predict the optimal point with the highest desirability, which was determined to be 46.73 g of ice plant paste and 20.23 g of egg. The response model was used to generate a graphical optimization and perturbation plot, as shown in Figure 5, with the characteristics of dependent variables presented in the overlay plot of Figure 5. The resulting sensory response values of the fresh ice plant paste pasta prepared using the optimal mixing ratio were 6.01 for color, 5.32 for flavor, 5.51 for taste, 5.80 for texture, 5.74 for appearance, and 5.79 for overall preference.

## 4. Discussion

Pasta is a staple food that is widely consumed due to its palatability, ease of preparation, low cost, and long shelf life [50]. The increasing global consumption of pasta presents an ideal opportunity to deliver functional ingredients and extra nutrients lacking in the diet [51]. Traditional pasta lacks dietary fiber and some micro nutrients, which can be addressed by fortifying pasta with functional ingredients. Some pasta production processes involve replacing refined durum wheat with other cereals, pseudocereals, and whole-grain cereals to create novel pastas [52]. Adding specific ingredients to pasta can deliver benefits to consumers, such as a more balanced amino acid content and increased antioxidant capacity. For example, buckwheat-enriched pasta has a higher antioxidant capacity than control pasta [53], and whole-meal pasta can increase satiety. Research has shown conflicting results with respect to the effect of fiber content on satiety and glycemia [54]. Some cereals also contain antinutritional factors, such as phytic acid, which may decrease the bioavailability of minerals and reduce their benefit. Fortification of pasta with functional ingredients is an important way to deliver more complete nutrition to consumers. However, the addition of these ingredients may affect the quality and acceptability of pasta and have conflicting effects on satiety and glycemia [55]. Further research is needed to fully understand the effects of pasta fortification.

Pasta with added vegetables contains a variety of essential nutrients, such as phytochemicals, fiber, vitamins, polyphenols, carotenoids, glucosinolates, and minerals [56]. Despite the health benefits of consuming these nutrients, customers often fail to meet the recommended daily intake of 400 g of fruits and vegetables as per World Health Organization guidelines (Marinelli et al., 2018) [57]. This could be due to unpleasant flavors, such as bitterness, that reduce consumer acceptability (Meengs, Roe, and Rolls, 2012) [58]. Most vegetable enrichment efforts focus on the high antioxidant properties of these ingredients. Antioxidants such as phenolic compounds, vitamin C (ascorbic acid), vitamin E (tocopherols), carotenoids, phytosterols, isoflavones, and organosulfur provide various health benefits, such as scavenging of free radicals, a reduction in carcinogenesis, binding of toxins and carcinogens in the intestines, reduced cholesterol absorption, and immune system enhancement [59]. Dietary fiber also helps to trap harmful toxins in the intestines, lower serum LDL cholesterol, decrease fat absorption, and reduce the risk of cardiovascular disease, in addition to acting as a prebiotic for beneficial bacteria in the intestines [60]. Traditional pasta, however, is lacking in essential minerals due to the conventional milling process of durum wheat, which causes a significant reduction in minerals such as zinc and iron [61]. Cooking can also result in a loss of minerals such as potassium, iron, zinc, and copper, although an increase in calcium was observed in cooked pasta [62]. Adding certain vegetables to pasta can improve its nutritional value; for example, the addition of 3% moringa leaf powder to wheat semolina-pearl millet flour improved the zinc and iron contents of pasta [63]. The same study also showed an increase in calcium content when carrot powder and mango peel powder were used to substitute raw materials. However, the bioavailability of minerals from functional pasta is questionable due to the high phenolic content in vegetables, which may form insoluble complexes with minerals in the gastrointestinal tract and reduce their absorption [64]. Further research is needed to understand the bioavailability of minerals from functional pasta.

Vegetable supplementation in pasta has been evaluated, but most studies only use small amounts of vegetables to maintain overall acceptability. While pasta enriched with functional ingredients may have superior nutritional properties compared to traditional pasta, it may negatively impact technical and sensory qualities and be rejected by consumers [58]. The cooking quality of fortified pasta depends on the level of substitution, with lower levels resulting in reduced effects [65]. The impact of fiber on pasta texture also depends on the level of substitution, with higher fiber concentrations leading to lower texture quality [66]. To maximize quality impact and achieve consumer acceptability, the recommended level of functional ingredient replacement is less than 10% [65]. Adding carrot, spinach, tomato, and beetroot puree improved the shape and texture of pasta, increasing its carotene, lycopene, and betalain contents [67]. However, carrot powder had a negative impact on the sensory and culinary qualities of pasta [68]. Similarly, adding 10–15% tomato skin flour to spaghetti reduced its overall sensory quality [69]. Further research is needed to fully understand the effects of different forms of vegetable supplementation on the culinary, sensory, and nutritional qualities of pasta. Adding vegetable ingredients may also lead to unpleasant flavors and reduced acceptability [70]. For example, the addition of 5–8% moringa leaves caused a leafy and bitter taste, reducing consumer receptivity [63]. However, the supplementation of semolina-based fresh pasta with spirulina had a positive effect on nutritional quality in terms of protein and amino acid content, and increased nutritional value of fresh pasta was achieved with good consumer acceptance. Incorporating functional ingredients into pasta holds great potential for improving its nutritional value while still maintaining consumer acceptance [71]. Vegetables can provide beneficial bioactive compounds and increase the antioxidant capacity of pasta. The increased fiber content provided many functional ingredients can also bring additional health benefits. When choosing functional ingredients, it is important to consider their composition, including high amounts of bonded phenolic compounds and fewer water-soluble nutrients. To avoid a negative impact on sensory quality, low levels of fortification are recommended. Adding puree or liquid functional ingredients has been shown to result in a higher-quality end products. Therefore, a combination of both puree and paste forms could be considered as a way to produce high-quality, nutritious pasta.

## 5. Conclusions

In this study, we examined the components and minerals present in ice plant and analyzed their ability to inhibit the activity of starch-degrading enzymes, specifically focusing on D-pinitol and inositol. Additionally, we conducted an amino acid analysis to identify the presence of 20 essential amino acids. The findings of our study demonstrate that incorporating ice plant into semolina-based fresh pasta had a positive impact on its nutritional quality, particularly in terms of pinitol, inositol, and amino acid contents. Moreover, among the various observed physical and chemical characteristics, color played a significant role in the resulting product.

The cooking characteristics of the supplemented fresh pasta were similar to those of regular fresh pasta, except when ice plant paste was used as a replacement. Sensory evaluations indicated that samples containing 45 g of ice plant paste and 20 g of egg received the highest scores among all the supplemented fresh pasta samples. Developing a fortified pasta product with desirable quality poses a challenge, and it is crucial to accurately determine the proportions of individual additives based on consumer preferences. The inclusion of ice plant significantly enhanced the nutritional value of fresh pasta and was well-received by consumers in this study. However, future research should focus on sensory analysis to improve the overall sensory characteristics of pasta when ice plant is added under different conditions, thereby providing a higher nutritional value tailored to consumer acceptance. Furthermore, to explore additional health benefits of incorporating ice plant into food and develop fresh pasta that offers enhanced health advantages, further studies should investigate the elemental composition of ice-plant-incorporated pasta without excess water and other ingredients, particularly those with high levels of antioxidant activity.

## Figures and Tables

**Figure 1 foods-12-02482-f001:**
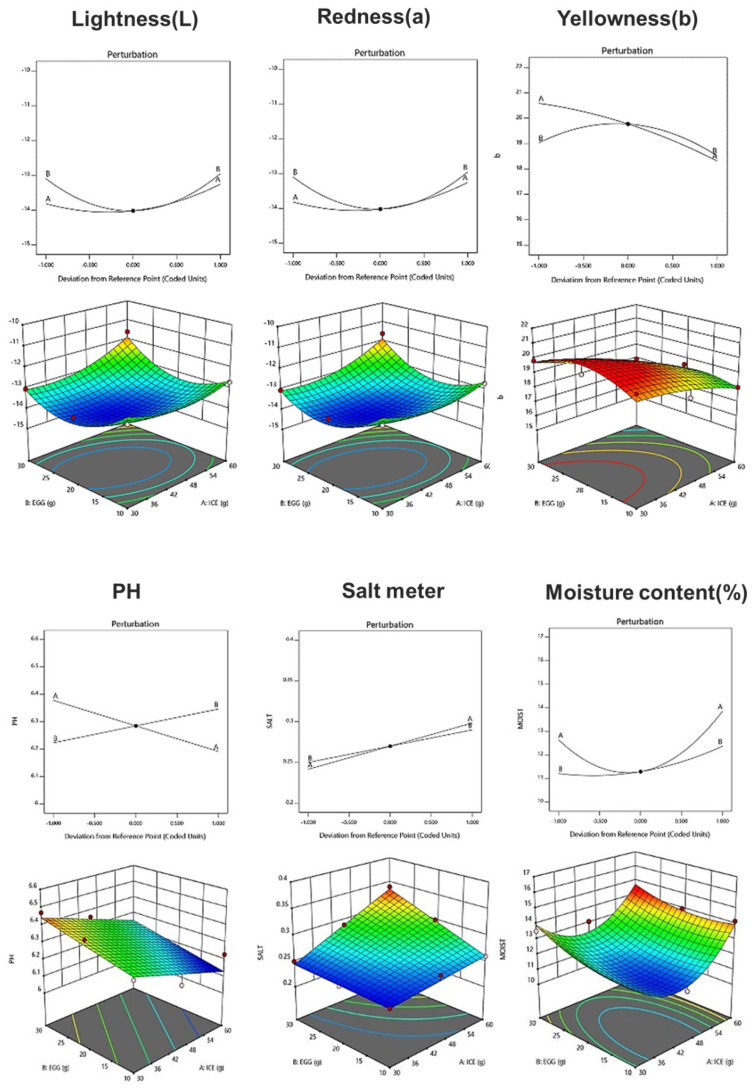
Perturbation plot showing the effect of ice plant (*M. crystallinum* L.) paste (A) and whole eggs (B) on the physicochemical properties and texture of the pasta.

**Figure 2 foods-12-02482-f002:**
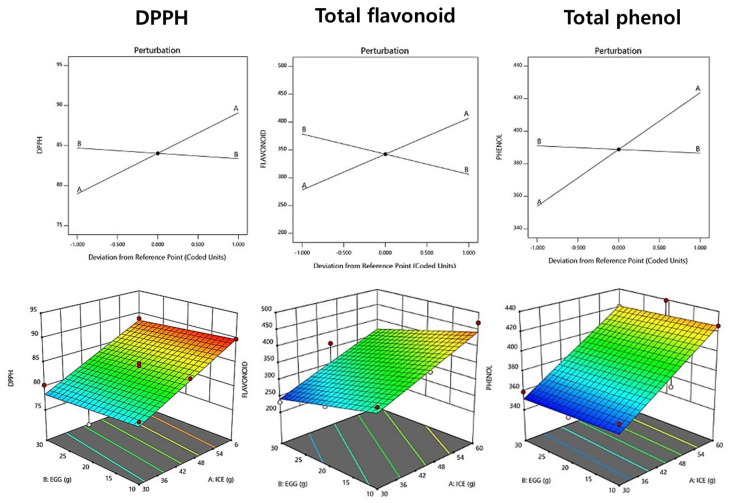
Perturbation plot showing the effect of ice plant (*M. crystallinum* L.) paste (A) and whole eggs (B) on starch-degrading enzyme inhibition and antioxidant activity of the pasta.

**Figure 3 foods-12-02482-f003:**
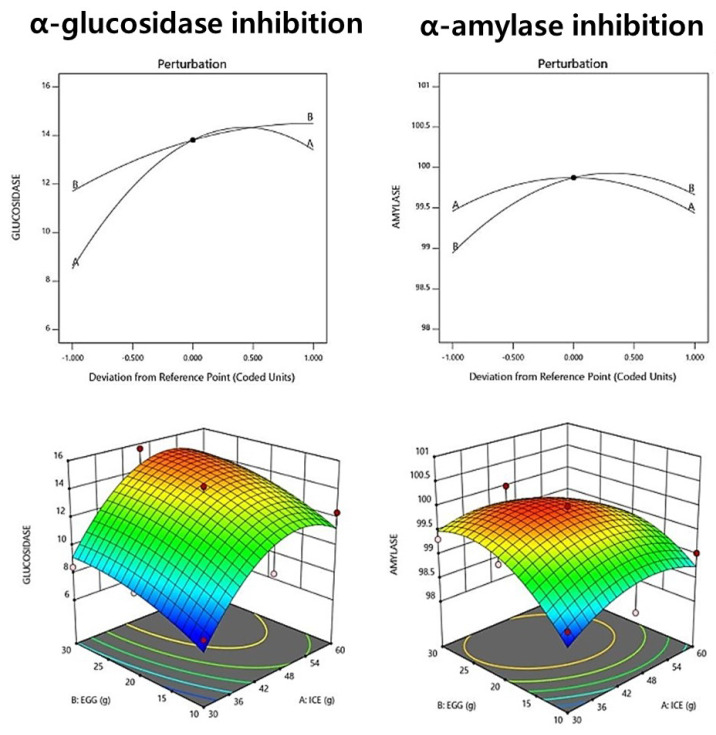
Perturbation plot showing the effect of ice plant (*M. crystallinum* L.) paste (A) and whole eggs (B) on starch-degrading enzyme inhibition and antioxidant activity of the pasta.

**Figure 4 foods-12-02482-f004:**
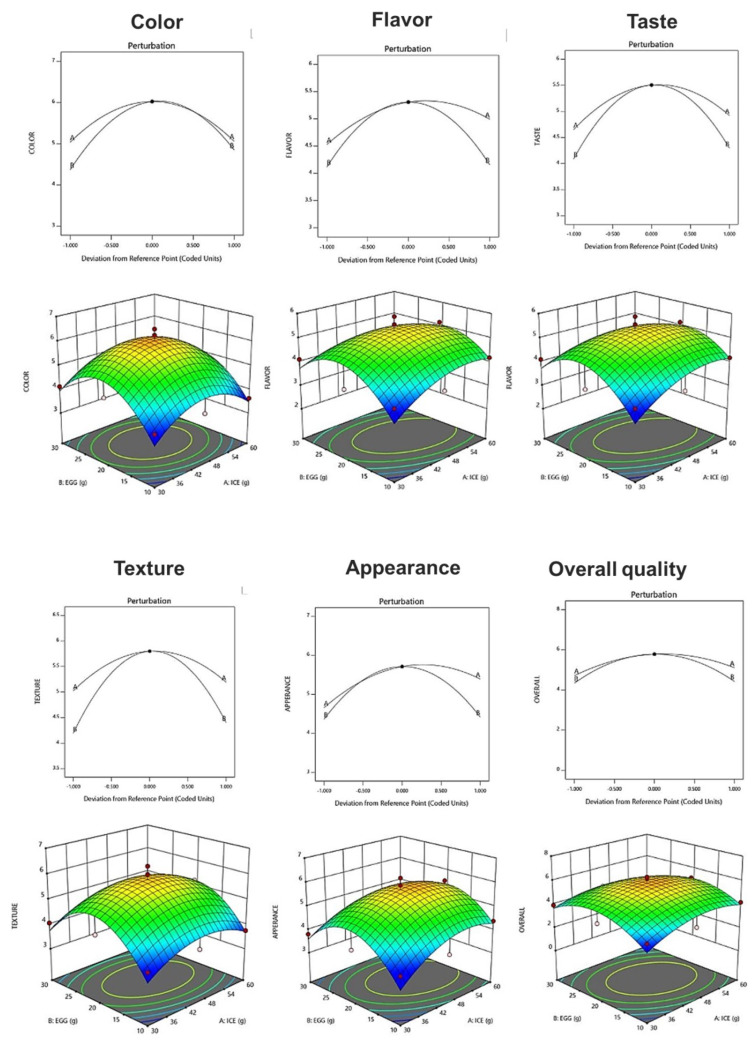
Perturbation plot showing the effect of ice plant (M. crystallinum L.) paste (A) and whole eggs (B) on starch-degrading enzyme inhibition and antioxidant activity of the pasta.

**Figure 5 foods-12-02482-f005:**
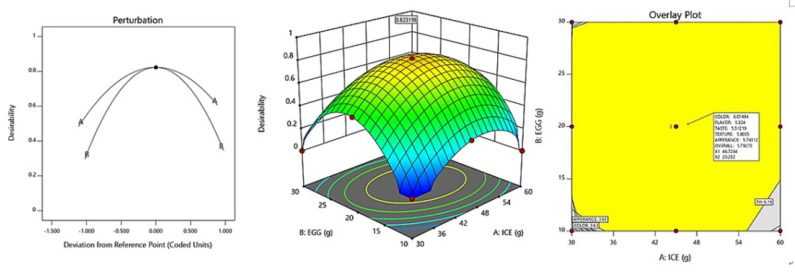
Perturbation plot and response surface plot of the effect of ice plant paste (A) and whole egg (B) on optimization of pasta with Ice plant.

**Table 1 foods-12-02482-t001:** Experimental design for pasta with ice plant paste.

No.	Variable Level			
	Ice Plant Raw Paste (g)	Egg (g)	Semolina Flour (g)	Salt (g)
1	60.00	30.00	100.00	2.00
2	30.00	10.00		
3	45.00	10.00		
4	60.00	10.00		
5	30.00	30.00		
6	30.00	20.00		
7	45.00	30.00		
8	60.00	20.00		
9	45.00	20.00		
10	45.00	20.00		

**Table 2 foods-12-02482-t002:** Analysis of proximate components each parts of ice plants. Mean ± S.D. (g/100 g).

Part					
	Moisture	Crude Protein	Crude Lipid	Crude Ash	Carbohydrate
Cotyledon	97.67 ± 0.01 ^c^ ***	0.51 ± 0.05 ^a^ ***	0.07 ± 0.02 ^b^ ***	0.87 ± 0.01 ^a^ ***	0.87 ± 0.05 ^a^ ***
Stem	90.60 ± 0.24 ^a^ ***	0.78 ± 0.03 ^b^ ***	0.03 ± 0.12 ^a^ ***	1.99 ± 0.46 ^c^ ***	6.60 ± 0.26 ^c^ ***
Shoot	95.89 ± 0.75 ^b^ ***	1.09 ± 0.80 ^c^ ***	0.13 ± 0.01 ^a^ ***	1.65 ± 0.06 ^b^ ***	1.35 ± 0.13 ^b^ ***

*** *p* < 0.001. All values are expressed as mean ± S.D. of triplicate determinations (*n* = 3). Values followed by different letters (^a–c^) means in a row when followed by different superscripts Duncan’s multiple range test.

**Table 3 foods-12-02482-t003:** Contents of raw minerals in different parts of ice plant. Mean ± S.D. (mg/kg).

	Mineral	Cotyledon	Stem	Leaf	F Value
Macro mineral	Na	3084.67 ± 9.61 ^b^	2864.33 ± 82.73 ^a^	2998.33 ± 16.26 ^b^	285.14 (0.000) ***
	K	1047.33 ± 2.89 ^a^	2086.00 ± 44.64 ^c^	1592.33 ± 27.03 ^b^	35.11 (0.000) ***
	Mg	108.33 ± 1.16 ^a^	174.00 ± 7.21 ^b^	441.00 ± 7.94 ^c^	889.33 (0.000) ***
	Ca	275.00 ± 8.00 ^a^	1971.33 ± 151.01 ^c^	988.00 ± 5.29 ^b^	15.41 (0.004) **
Micro mineral	Fe	1.87 ± 0.21 ^a^	3.60 ± 0.72 ^b^	5.23 ± 0.40 ^c^	2401.74 (0.000) ***
	Cu	0.14 ± 0.00 ^a^	0.25 ± 0.35 ^a^	1.66 ± 0.61 ^b^	8748.0 (0.000) ***
	Zn	2.10 ± 0.00 ^a^	7.53 ± 0.59 ^c^	6.27 ± 0.12 ^b^	203.92 (0.000) ***
	Mn	1.17 ± 0.06 ^a^	6.57 ± 0.06 ^a^	1.17 ± 0.06 ^b^	17.28 (0.003) **

** *p* < 0.01,*** *p* < 0.001. All values are expressed as mean ± S.D. of triplicate determinations (*n* = 3). Values followed by different letters (^a–c^) means in a row when followed by different superscripts Duncan’s multiple range test.

**Table 4 foods-12-02482-t004:** Pinitol and inositol values from different parts of ice plant. Mean ± S.D. (mg/100 g).

Component	Sample	Analysis Value	RSD (%)	F Value (*p*)
Pinitols	Cotyledon	4.18 ± 0.27 ^a^	6.42	45,875.58 (0.000) ***
	Stem	263.76 ± 1.43 ^c^	0.54	
	Shoot	138.14 ± 1.09 ^b^	0.79	
Inositols	Cotyledon	4.36 ± 0.24 ^a^	5.61	1772.46 (0.000) ***
	Stem	44.45 ± 1.44 ^c^	3.25	
	Shoot	10.53 ± 0.47 ^b^	4.45	

RSD = relative standard deviation. Values are expressed as mean ± S.D. of three replications. *** *p* < 0.001.

**Table 5 foods-12-02482-t005:** Amino acid content of different parts of ice plant. Mean ± S.D. (mg/100 g).

Amino Acid Type	Cotyledon	Stem	Leaf
Aspartic acid	1537.44	1619.19	1327.44
Glutamic acid	2637.55	2744.47	1257.64
Asparagine	897.08	1469.73	496.02
Serine	1508.17	1685.61	651.92
Glutamine	7011.85	5350.20	2763.16
Histidine	983.98	1049.26	401.72
Glycine	340.09	707.38	282.68
Threonine	1248.19	1570.88	691.45
Arginine	1476.09	2371.91	1253.82
Alanine	1508.86	1842.42	1040.36
GABA	293.74	157.99	376.94
Tyrosine	1501.83	2031.06	711.08
Valine	2101.76	1961.52	912.43
Methionine	524.21	884.61	220.74
Tryptophane	963.50	1102.91	387.61
Phenylalanine	2031.21	2237.81	844.81
Isoleucine	1307.47	1395.43	569.66
Leucine	2366.92	3373.33	1268.73
Lysine	2018.88	2392.21	953.48
Proline	775.00	815.68	467.90

**Table 6 foods-12-02482-t006:** Total dietary fiber content of different parts of the ice plant. Mean ± S.D. (g/100 g).

	Cotyledon	Stem	Leaf	F Value ^1)^
Content	1.44 ± 0.19 ^a^	3.84 ± 0.16 ^c^	2.35 ± 0.20 ^b^	49,794.875 ***

^1)^ *** *p* < 0.001. means in a row when followed by different superscripts Duncan’s multiple range test. Values are expressed as mean ± S.D. of three replications.

**Table 7 foods-12-02482-t007:** Inhibitory effects in different parts of ice plant on α-amylase activity and α-glucosidase activity. Mean ± S.D. (mg/mL).

Content	Inhibitory Effects	
	α-Glucosidase Activity (%)	α-Amylase Activity (%)
Control (acarbose)	43.63 ± 0.33	83.61 ± 1.60
Cotyledon	5.99 ± 0.53 ^a^	17.54 ± 1.70 ^a^
Stem	25.45 ± 0.55 ^b^	31.91 ± 3.90 ^c^
Shoot	5.52 ± 0.59 ^a^	23.04 ± 1.55 ^b^
F value	3791.43 (0.000) ***	478.06 (0.000) ***

Values are expressed as means ± S.D. of three independent experiments. *** *p* < 0.001.

**Table 8 foods-12-02482-t008:** Physicochemical properties of the pasta prepared with ice plant paste. Mean ± S.D.

No	Factor		Response (Mean ± S.D)					
	Ice Plant Paste (g)	Egg (g)	L ^1)^	A ^2)^	B ^3)^	pH	Salt Meter (%)	Moisture Content (%)
1	60.00	30.00	30.55 ± 0.68	−11.61 ± 0.14	16.38 ± 0.28	6.23 ± 0.30	0.34 ± 0.01	14.41 ± 0.06
2	30.00	10.00	36.87 ± 0.53	−12.88 ± 0.16	19.93 ± 0.49	6.30 ± 0.35	0.24 ± 0.01	12.45 ± 0.05
3	45.00	10.00	34.05 ± 1.01	−12.73 ± 0.37	18.48 ± 0.63	6.16 ± 0.12	0.26 ± 0.00	10.91 ± 0.01
4	60.00	10.00	32.89 ± 0.35	−12.71 ± 0.87	18.01 ± 0.17	6.00 ± 0.30	0.26 ± 0.01	14.18 ± 0.23
5	30.00	30.00	33.55 ± 2.40	−13.05 ± 0.18	19.83 ± 0.16	6.47 ± 0.31	0.25 ± 0.00	13.53 ± 0.03
6	30.00	20.00	34.08 ± 0.44	−13.53 ± 0.26	19.99 ± 0.69	6.41 ± 0.03	0.24 ± 0.01	12.95 ± 0.03
7	45.00	30.00	31.33 ± 1.17	−13.24 ± 0.41	18.70 ± 0.60	6.36 ± 0.17	0.29 ± 0.00	13.09 ± 0.30
8	60.00	20.00	34.25 ± 0.44	−13.46 ± 0.19	18.55 ± 0.32	6.16 ± 0.12	0.30 ± 0.01	13.97 ± 0.15
9	45.00	20.00	36.19 ± 0.82	−14.08 ± 0.02	19.94 ± 0.08	6.27 ± 0.10	0.26 ± 0.00	11.07 ± 0.06
10	45.00	20.00	36.11 ± 0.43	−14.05 ± 0.78	19.96 ± 0.03	6.26 ± 0.38	0.26 ± 0.00	11.08 ± 0.07

^1)^ L: lightness (white +100 ↔ 0 black); ^2)^ A: redness (red +60 ↔ −60 green); ^3)^ B: yellowness (yellow +60 ↔ −60 blue).

**Table 9 foods-12-02482-t009:** Analysis of the predicted model equation for the physicochemical properties of pasta prepared with ice plant paste.

Response	Model	Mean ± S.D.	R^2 1)^	F Value	Prob > F ^2)^	Polynomial Equation ^3)^
L	Linear	33.99 ± 2.06	0.51	967.93	0.0246 *	33.99 − 1.14A − 1.4B
a	Linear	−13.13 ± 0.73	0.89	13.07	0.018 *	−14.02 + 0.28A + 0.28B + 0.3175AB + 0.3175A^2^ + 0.9957B^2^
b	Linear	18.97 ± 1.19	0.91	6.11	0.03 *	19.77 − 1.14A − 0.255B − 0.3875AB − 0.3243A^2^ − 1B^2^
pH	Linear	6.28 ± 0.10	0.8	14	0.0036 **	6.29 − 0.933A + 0.0617B
salinity	Linear	0.27 ± 0.03	0.96	20.51	0.004 **	0.27 + 0.0283A + 0.02B + 0.075AB
Moisture content (%)	Quadratic	12.76 ± 1.34	0.84	7448.77	0.0085 **	11.29 + 0.605A + 0.50817 − 0.2125AB + 1.96A^2^ + 0.5007B^2^

^1)^ 0 ≦ R^2^ ≦ 1, close to 1 indicates the regression line fits the model; ^2)^ * *p* < 0.05, ** *p* < 0.01; ^3)^ A: egg; B: Ice plant paste.

**Table 10 foods-12-02482-t010:** Antioxidant activities of pasta with ice plant paste. Mean ± S.D.

Sample No.	Ice Plant Paste (g)	Egg (g)	Response		
			DPPH Radical Scavenging Activity (%)	Total Flavonoids (mg QE/g)	Total Phenols (mg GAE/g)
1	60.00	30.00	88.81 ± 5.78	359.42 ± 68.66	420.88 ± 20.21
2	30.00	10.00	80.51 ± 5.33	330.25 ± 27.84	365.88 ± 7.30
3	45.00	10.00	85.11 ± 1.18	373.58 ± 7.64	381.86 ± 8.91
4	60.00	10.00	89.80 ± 2.27	470.25 ± 30.00	426.05 ± 7.50
5	30.00	30.00	80.32 ± 4.48	231.08 ± 14.65	358.87 ± 12.28
6	30.00	20.00	75.90 ± 3.64	275.25 ± 71.46	351.88 ± 8.62
7	45.00	30.00	82.31 ± 1.56	366.08 ± 73.16	380.38 ± 13.29
8	60.00	20.00	88.45 ± 1.65	393.58 ± 143.62	439.22 ± 27.08
9	45.00	20.00	84.86 ± 4.90	311.92 ± 10.41	381.38 ± 4.54
10	45.00	20.00	84.89 ± 2.93	311.75 ± 10.23	381.05 ± 4.29

**Table 11 foods-12-02482-t011:** Analysis of the predicted model equation for antioxidant activity of pasta prepared with ice plant paste. Mean ± S.D.

Response	Model	Mean ± S.D.	R^2 1)^	F-value	Prob > F ^2)^	Polynomial Equation ^3)^
DPPH radical scavenging activity (%)	Linear	84.04 ± 4.38	0.903	32.39	0.0003 ***	84.04 + 5.05A − 0.6633B
Total flavonoids (mg QE/g)	Linear	342.32 ± 66.42	0.826	16.64	0.0022 ***	342.32 + 64.44A − 36.25B
Total phenols (mg GAE/g)	Linear	388.75 ± 29.77	0.921	40.84	0.0001 ***	388.75 + 34.92A − 2.28B

^1)^ 0 ≦ R^2^ ≦ 1, close to 1 indicates the regression line fits the model; ^2)^ *** *p* < 0.001; ^3)^ A: egg; B: Ice plant paste.

**Table 12 foods-12-02482-t012:** Starch-degrading enzyme activity inhibition of pasta with raw ice plant paste. Mean ± S.D.

No.	Ice Plant Paste (g)	Egg (g)	Response	
			α-Glucosidase Inhibitory Activity (%)	α-Amylase Inhibitory Activity (%)
1	60.00	30.00	13.72 ± 2025	89.81 ± 1.70
2	30.00	10.00	7.25 ± 4.26	68.59 ± 1.85
3	45.00	10.00	9.76 ± 1.22	78.35 ± 0.75
4	60.00	10.00	12.38 ± 1.51	89.03 ± 0.50
5	30.00	30.00	8.43 ± 0.82	69.32 ± 1.22
6	30.00	20.00	8.44 ± 1.14	69.32 ± 0.43
7	45.00	30.00	15.64 ± 1.84	89.90 ± 0.20
8	60.00	20.00	12.68 ± 1.83	99.02 ± 0.20
9	45.00	20.00	14.18 ± 3.58	89.90 ± 0.84
10	45.00	20.00	14.24 ± 3.56	89.90 ± 0.86

**Table 13 foods-12-02482-t013:** Analysis of the predicted model equation for antidiabetic activities of pasta with ice plant paste. Mean ± S.D.

Response	Model	Mean ± S.D.	R^2 1)^	F-Value	Prob > F ^2)^	Polynomial Equation ^3)^
**α-Glucosidase** inhibitory activity (%)	Quadratic	11.67 ± 2.95	0.921	5.24	0.0384 *	13.81 + 2.44A + 1.4B + 0.04AB − 2.85A^2^ − 0.7129B^2^
**α-Amylase** inhibitory activity (%)	Quadratic	83.27 ± 0.595	0.861	3.73	0.123	99.87 − 0.0117A + 0.36B − 0.2375AB − 0.4271A^2^ − 0.5721B^2^

^1)^ 0 ≦ R^2^ ≦ 1, close to 1 indicates the regression line fits the model; ^2)^ * *p* < 0.05; ^3)^ A: egg; B: Ice plant paste.

**Table 14 foods-12-02482-t014:** Sensory properties of pasta with ice plant paste. Mean ± S.D.

No.	Ice Plant Paste (g)	Egg (g)	Response					
			Color	Flavor	Taste	Texture	Appearance	Overall Quality
1	60.00	30.00	3.69 ± 1.50	3.31 ± 1.20	3.31 ± 1.14	3.63 ± 1.36	3.63 ± 1.31	3.19 ± 1.42
2	30.00	10.00	3.75 ± 1.30	3.56 ± 1.26	3.50 ± 1.10	3.69 ± 1.20	4.00 ± 1.27	3.63 ± 1.31
3	45.00	10.00	3.75 ± 0.86	3.50 ± 1.37	3.63 ± 1.03	3.75 ± 1.00	3.69 ± 1.08	3.44 ± 0.81
4	60.00	10.00	3.63 ± 1.26	4.19 ± 1.67	3.69 ± 0.87	4.13 ± 1.26	4.38 ± 1.20	4.19 ± 1.05
5	30.00	30.00	4.13 ± 0.96	4.12 ± 1.09	3.88 ± 1.20	4.06 ± 1.12	3.81 ± 0.91	3.94 ± 1.12
6	30.00	20.00	4.38 ± 0.89	3.56 ± 0.63	3.94 ± 1.10	4.31 ± 0.95	3.88 ± 0.96	3.75 ± 0.86
7	45.00	30.00	4.75 ± 0.93	3.94 ± 0.68	4.25 ± 0.86	4.13 ± 1.03	4.44 ± 0.89	4.38 ± 0.72
8	60.00	20.00	5.00 ± 1.21	5.13 ± 1.03	5.13 ± 0.99	5.19 ± 1.28	5.52 ± 0.93	5.13 ± 0.72
9	45.00	20.00	6.25 ± 0.68	5.56 ± 0.89	5.63 ± 1.36	6.00 ± 0.89	5.88 ± 0.81	6.19 ± 0.66
10	45.00	20.00	6.50 ± 0.52	5.88 ± 0.89	5.88 ± 0.96	6.31 ± 0.79	6.19 ± 0.75	6.31 ± 0.48

**Table 15 foods-12-02482-t015:** Analysis of the predicted model equation for the sensory properties of pasta with ice plant (*M. crystallinum* L.) paste.

Response	Model	Mean ± S.D.	R^2^	F Value	Prob > F	Polynomial Equation
Color	Quadratic	4.28 ± 0.92	0.862	11.99	0.0204	6.02 + 0.01A + 0.24 − 0.08AB − 0.9793A^2^ − 1.42B^2^
Flavor	Quadratic	4.28 ± 0.93	0.701	3.95	0.1131	5.31 + 0.2317A + 0.02B − 0.36AB − 0.5464A^2^ − 1.17B^2^
Taste	Quadratic	4.28 ± 0.93	0.840	9.99	0.0278	5.51 + 0.135A + 0.1033B − 0.19AB − 0.72A^2^ − 1.31B^2^
Texture	Quadratic	4.48 ± 0.99	0.850	11.06	0.0234	5.8 + 0.085A + 0.105B − 0.1225AB − 0.6921A^2^ − 1.5B^2^
Appearance	Quadratic	4.51 ± 0.99	0.801	6.87	0.0500	5.71 + 0.3667A + 0.0283B − 0.23AB − 0.6864A^2^ − 1.32B^2^
Overall quality	Quadratic	4.42 ± 1.11	0.745	5.36	0.0738	5.78 + 0.1983A + 0.0417B − 0.3275AB − 0.8743A^2^ − 1.4B^2^

## Data Availability

The data presented in this study are available on request from the corresponding author.

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
