# Peer review of "Optimization of Nutrient-Rich Ice Plant (Mesembryanthemum crystallinum L.) Paste Fresh Noodle Pasta Using Response Surface Methodology"

_foods, 2023, doi:10.3390/foods12132482_

Round 1
Reviewer 1 Report
The idea is interesting. Generally, article has not prepared good enough from the view of linguistic. Additionally, the modelling structure should be improved.
Lines 21-22: The optimal conditions were found to be 46.73 g Ice plant paste and 20.23 g 21 eggs, providing a method for developing raw pasta with enhanced health benefits from Ice plant paste without excessive water and other ingredients. This is not clear! How much amount of producing pasta? This should be revised.
Keywords: Keywords and title should be comprised of different words.
Line 99: The carbohydrate content is the value excluding moisture, ash, crude protein, and crude fat from 100. What is 100? This sentence should be revised.
Equation 1: It does not look like scientific. All equations should be prepared more professional. Please revise all throughout the paper.
Lines 176-179, 258-259: Central composite or Box-Behnken which one is true!?
Table 1: What is standart no.? Coded values should be also shown here.
Lines 263, 286, 316, 357, 422, 473: probability value is 0.05 or 0.001? Please be consistent!
Table 2: superscript (***) where is it!?
Figure 1. A pertubation plot showing the effect of………. What is pertubation? Line 486. Please check all article.
Line 490: 1) L: lightness (white +100 ↔ 0 black).2) a: redness (red +60 ↔ −60 green).3) b: yellowness (yellow 490 +60 ↔ −60 blue). Are you sure?
Table 13. α-amylase inhibitory activity. Please assess results of quadratic model? What is the meaning of p = 0.123. All tables and results should be evaluated from this perspective.
Conclusions: Conclusion part should be revised.
article has not prepared good enough from the view of linguistic
Author Response
Reviewer 1 Amendment Request
Q1: Lines 21-22.
A: Revised part has been highlighted in red
Q2: Keywords:
A: Shall we change the title?? Please mention keywords to change
Q3: Line 99
A: Revised part has been highlighted in red
Q4: Equation 1: It does not look like scientific. All equations should be prepared more professional. Please revise all throughout the paper
A: I did not understand. It's not an equation, it's just a way of expressing capacity in words.
Q5: Lines 176-179, 258-259
A: Revised part has been highlighted in red
Q6: Table 1
A: Revised part has been highlighted in red
Q7: Lines 263, 286, 316, 357, 422, 473
A: Revised part has been highlighted in red
Q8: Table 2: superscript (***)
A: Revised part has been highlighted in red
Q9: Figure 1. pertubation
Line 486.
A: In analysis, perturbation theory is a method used to approximate solutions to problems that are difficult or impossible to solve exactly. It involves introducing small variations or perturbations into a known solution to simplify the problem.
Q10: Line 490: 1) L: lightness (white +100 ↔ 0 black).2) a: redness (red +60 ↔ −60 green).3) b: yellowness (yellow +60 ↔ −60 blue). Are you sure
A: These are the numbers that usually appear on a colorimeter.
Q11: Table 13. quadratic model
What is the meaning of p = 0.123. All tables and results should be evaluated from this perspective.
A: The relationship between ice plant paste and egg factors in the context of an interacting quadratic model
Q12: Conclusions
A: It was revised by adding colors from line 562.

Reviewer 2 Report
Introduction
The authors presented the nutritional uniqueness of the ice plant and the importance of the paste as a food product consumed by many populations. The authors also presented recent developments on pasta production, mainly fortified to increase nutritional value. At the end of the chapter, they clearly stated the purpose and scope of they work.
I recommend the literature to be used in the introduction, results and discussion of the results: 10.3390/molecules27020355, 10.3390/foods11162456
Materials and Methods
2.9.1.Color Measurement - please write the measurement parameters and the device on which the color of the samples was measured.
2.9.2. PH Measurement - sugar/saccharose measurement information should be removed from this subsection
2.9.3. Salt Meter Measurement- please correct the title of the subsection and provide the name of the measuring device and the method by which the measurement was made.
2.9.5.DPPH, Total phenol and Total flavonoids Antioxidant - a single-extraction was performed? How do the authors know that it was enough to extract all the compounds?
2.9.6. Starch degrading enzyme activity inhibition of raw pasta with Ice plant paste - this subsection should be eliminated and the information should be moved to sections 2.7 and 2.8.
3.Results - please do not repeat the numerical information given in the tables - especially this applies to subsection 3.1. Please correct this.
Discussion
I have no comments to this part of article.
Conclusions
line 668-672- this text should be removed because it's a repetition of information, and scope of research already given earlier.
Moreover, the conclusions are sparse and should be expanded, especially since the authors made so many determinations, and RSM.
Author Response
Reviewer 2 Amendment Request
Q1: 10.3390/molecules27020355, 10.3390/foods11162456
A: The above paper has been revised by adding to the discussion.
Q2: 2.9.1.Color Measurement
A: The manufacturer, country of manufacture, and machine name of the machine were indicated and corrected in blue letters.
Q3: 2.9.2. PH Measurement
A: The manufacturer, country of manufacture, and machine name of the machine were indicated and corrected in blue letters.
Q4: 2.9.3. Salt Meter Measurement
A: The manufacturer, country of manufacture, and machine name of the machine were indicated and corrected in blue letters.
Q5: 2.9.5.DPPH, Total phenol and Total flavonoids Antioxidant - a single-extraction was performed? How do the authors know that it was enough to extract all the compounds?
A: Ethanol single extraction was performed. I thought that it should be extracted with alcohol in order to consume it by previous research. I will try to experiment again with various solvents in the future.
Q6: 2.9.6. Starch degrading enzyme activity inhibition of raw pasta with Ice plant paste - this subsection should be eliminated and the information should be moved to sections 2.7 and 2.8.
A: It was deleted, relocated, and marked with blue letters.
Q7: 3.Results - please do not repeat the numerical information given in the tables - especially this applies to subsection 3.1. Please correct this.
A: Deleted and revised.
Q8: Discussion.
A: Revised in blue text.
Q9: Conclusions
A: Revised in blue text.
Q10: line 668-672- this text should be removed
A: Deleted and revised.

Round 2
Reviewer 1 Report
The paper can be published.
Minor editing of English language required.
Reviewer 2 Report
The authors have addressed all comments and feedback.
They expanded the description of the methodology so that it could be reproduced. The text presenting the results no longer repeats the data in the tables. The authors have further expanded the discussion of the results. Most importantly, they removed repetitive information in the conclusion chapter and expanded it significantly.
I have no further comments.